# Cultural and Practical Barriers to Seeking Help for Intimate Partner Violence Among Korean Immigrants: Exploring Gender and Age Differences

**DOI:** 10.3390/ijerph21111508

**Published:** 2024-11-13

**Authors:** Soon Cho, Y. Joon Choi, Jeong-Yeob Han, Hanyoung Kim, Stephen T. Fife

**Affiliations:** 1Department of Community, Family, and Addiction Sciences, Texas Tech University, Lubbock, TX 79415, USA; stephen.fife@ttu.edu; 2School of Social Work, Andrew Young School of Policy Studies, Georgia State University, Atlanta, GA 30302, USA; ychoi54@gsu.edu; 3Department of Advertising and Public Relations, University of Georgia, Athens, GA 30602, USA; jeonghan@uga.edu; 4Department of Integrated Strategic Communication, University of Kentucky, Lexington, KY 40506, USA; hanyoung.kim@uky.edu

**Keywords:** Confucian culture, domestic violence, family violence, focus group, immigrant women, intimate partner violence, Korean American, Korean culture, Korean immigrant

## Abstract

Immigrants in the United States, including Korean immigrants, are more vulnerable to intimate partner violence (IPV), yet they are less likely to seek help than non-immigrants. This qualitative study sought to understand barriers to seeking help for IPV among Korean immigrants and to explore age and gender differences in Korean immigrants’ understanding of these barriers. We conducted four focus groups with 38 adults grouped by age and gender. Thematic analysis was employed to understand Korean immigrants’ perceptions of IPV and barriers to help-seeking. As a result, four prominent themes emerged: (1) differential understanding of IPV, (2) Confucian cultural influences on IPV, (3) cultural barriers to help-seeking, and (4) practical barriers to help-seeking. Women participants showed an in-depth understanding of IPV, recognizing various forms beyond physical violence within the immigrant social context. Younger participants highlighted the intergenerational transmission of IPV. Cultural factors, rooted in Confucianism such as strict gender roles and women’s self-sacrifice, exacerbate IPV. Cultural barriers include shame culture, treating IPV as a private matter, and the emphasis on family unity. Practical barriers are linked to the challenges immigrants face. These insights illustrate the need for targeted IPV interventions tailored to distinct gender and age demographics within the community.

## 1. Introduction

Immigrant women in the US, including Korean immigrant women, face heightened vulnerability to intimate partner violence (IPV) [1,2,3]. To our knowledge, there is no national-level research on the prevalence and types of IPV among Korean immigrants. However, the IPV prevalence rate may be high based on community-based research for Korean immigrant women for the last several decades [3,4,5,6,7]. In the 1980s, although only a small fraction of the Asian American population was of Korean descent, Koreans accounted for 33% of cases of wife abuse in the Asian American battered women’s shelter in Los Angeles [8]. Among 150 Korean immigrant women, Song [9] found that 60% reported being physically abused, 20% of them reported being abused once a week, and 37% reported being abused once a month. The research conducted by Shin [6] found that at least one-third of Korean male respondents reported using physical force against their wives in the past 12 months, according to the study conducted in 1995. Though wives were also reported to use violence, husbands tended to be more violent. Further, husbands were more likely to initiate violence and inflict greater injury on their wives. Also, based on data from 2002, among first-generation Korean immigrants, Ahn [10] reported a significant rate of partner abuse, including physical violence and verbal aggression, in the past 12 months. Moreover, according to a 2007 study by Lee [3], approximately 30% of women in the study experienced physical assault by their male partners, and 85% of the assaulted women were physically injured in the past 12 months. In contrast, a 2012 study by Liles et al. [11] reported that physical assault and injury among Korean immigrant women were uncommon (2%) in the past year, with psychological aggression (27.4%) being the most prevalent form of abuse. There is some evidence that severe physical assaults have decreased over time since the 1980s and 1990s; however, psychological aggression continues to be prevalent, with the age group 18–39 having the highest prevalence of psychological aggression in the past year [11].

Barriers that abused women face in seeking help are exacerbated in immigrant communities. Unique barriers for immigrant survivors include immigration status, social isolation and lack of family support, cultural and religious barriers, language and financial barriers, limited awareness of available resources, limited use and perceived inefficacy of services, fear of being deported and separated from their children, and psychological barriers such as feelings of shame and stigma, etc. [1]. Additionally, immigrants can be subjected to a distinct form of abuse known as immigration abuse, where abusers exploit their partners’ dependence by threatening to withdraw sponsorship or instigate deportation, leaving them financially reliant without alternative support networks [12,13]. Immigration abuse also includes restricting the contact of women with their families in their native countries and preventing them from learning English, which may decrease their acculturation and independence [14]. In addition, cultural and religious values also contribute to IPV and to the shame of seeking help. These values are male-dominated family structure, rigid gender roles, ingrained patriarchal and hierarchal family systems, high value of family honor and privacy, and value of group welfare over personal needs [2,15].

Numerous cultural and psychosocial barriers experienced by abused immigrant women lead them to exhaust all informal resources, such as seeking help from religious leaders, before turning to formal service providers [16,17]. This reluctance to seek formal help for IPV among immigrant survivors due to cultural and practical barriers highlights the need for ethnic-specific interventions that address these obstacles and increase survivor access to services and resources. Effective strategies to enhance survivors’ access to services must incorporate the values and perspectives of their communities. However, there is little empirical research documenting the perspectives of Korean immigrants in the US on IPV and socio-cultural barriers to seeking help. Furthermore, research on gender and age differences in how Korean Americans perceive these barriers is especially lacking. Our team sought to address this gap. The purpose of our study was to explore Korean immigrants’ community members’ perceptions of IPV and cultural and practical barriers to seeking help, with a focus on age and gender differences. While we did not specifically recruit participants with lived experience of IPV for our study, this approach provided us with the opportunity to engage a broader segment of the community in the discussion that might not have occurred if we had asked participants directly about their experiences. As a result, the method is more aligned with cultural norms, which favor indirect inquiries, particularly in contexts where discussing personal experiences may impede participation. We employed a qualitative approach to gain a deep understanding of the perspectives and insights of this target population, aiming to inform culturally responsive interventions, targeting the entire Korean immigrant community to effectively change perceptions and attitudes toward IPV.

## 2. Materials and Methods

### 2.1. Setting and Design

To understand the complex dynamics of intimate partner violence (IPV) among Korean immigrants, we drew on intersectionality theory as a theoretical framework. In this approach, various forms of systems, such as gender, age, race, class, education level, language proficiency, cultural heritage, acculturation, and immigration status, are not separated from each other but interact and affect individuals in multiple ways [18]. An understanding of intersectionality is particularly important when analyzing immigrants’ perceptions about how culture and immigration context contribute to IPV and help-seeking behavior. The purpose of our design was to capture the nuanced experiences of IPV that are influenced by the intersection of gender, age, and culture within the Korean immigrant community. We incorporated qualitative methods that allowed participants to discuss their IPV-related experiences and perceptions within a culturally contextual framework.

### 2.2. Sample

Upon receiving approval from the Institutional Review Board at the primary investigator’s institution (University of Georgia, IRB ID #PROJECT00001676), four focus groups were conducted with Korean American immigrants. Thirty-eight adults were organized into four focus groups by age and gender: the younger female group (20–30 s, n = 10), the younger male group (20–30 s, n = 10), the older female group (50–60 s, n = 8), and the older male group (70–80 s, n = 10). All participants met the following criteria: self-identified as Korean or Korean American, spoke Korean, were 18 years or older, and lived in the state of Georgia.

Almost all participants were born in Korea (36 out of 38). Participants in the older groups tended to have longer average years of residency in the U.S., with 29.4 years for older females and 34.4 years for older males. The younger groups reported 15.3 years for younger females and 15.6 years for younger males. In most groups, participants were predominantly Protestant Christians (10 out of 10 younger females; 8 out of 10 younger males; 8 out of 10 older males), except for the older female group (3 out of 8), where four reported affiliations with the Catholic Church. Participants’ education levels ranged from less than high school to a master’s degree. Notably, older females tended to have higher education levels, with all eight having reached college or obtained a master’s degree. In contrast, only two out of ten older males had education beyond high school. Younger participants were more likely to have both Korean and English as their primary languages than older participants. Table 1 shows the sociodemographic characteristics of the study participants.

Participants were recruited through Korean churches, senior centers, and snowball sampling in a large metropolitan city in the Southeast. The focus groups were held in a Korean immigrant church, a meeting room at a Korean restaurant, and via Zoom due to COVID-19 restrictions. Four native Korean-speaking co-authors facilitated the sessions, with gender-matched facilitators for each group. The first three focus groups were audio-recorded with participants’ permission, while the last group was video-recorded via Zoom, also with participants’ permission. Each session lasted approximately 90 min, and participants received a $40 gift card for their time.

Before the focus groups commenced, the composition was carefully designed to create an environment conducive to honest discussions on sensitive topics such as IPV [19]. Participants were assured of their confidentiality, signed an informed consent form that described the study and completed a brief form inquiring about demographic characteristics. Additionally, participants were instructed not to reveal any identifiable information regarding the cases they discussed and were encouraged to speak to the extent they felt comfortable doing so. In this way, participants could freely express their views without having to worry about privacy or being exposed. Facilitators summarized and documented emerging themes in field notes after each focus group.

### 2.3. Measures

The facilitators introduced the purpose of the study and the procedures by defining IPV. Facilitators defined IPV as physical abuse, emotional abuse, sexual abuse, and economic abuse with examples. After explaining the definition, we asked the following overarching questions:Share some of the issues you’ve seen or heard within the Korean community regarding family violence, particularly against spouses or partners.What aspects of our culture may contribute to IPV?What do you think are some of the cultural barriers to seeking help (for themselves or others) when someone is in an abusive relationship?What do you think are some of the practical barriers to seeking help (for themselves or others) when someone is in an abusive relationship?

The facilitators also ensured participants were advised not to share their own personal experiences but rather to discuss what they know about family violence among Korean immigrants in the U.S. to avoid identifying anyone.

### 2.4. Analysis

The audio files from the focus group studies were transcribed from Korean to English by the authors. Two native Korean-speaking authors were responsible for translating the focus group discussions into English. Both authors independently translated the transcripts and then compared their versions in order to resolve any discrepancies and ensure accuracy. Thematic analysis was employed, utilizing open coding as outlined by Patton [20]. Initially, the authors reviewed the field notes and a total of 71 single-spaced pages of focus group transcripts to delineate units of analysis. Subsequently, the transcripts and field notes were examined again using inductive coding. This was followed by a comparative analysis of the codes, in which we compared codes within and across the participants from the different focus groups and organized them according to their similarities and differences. Themes were then identified from prominent codes and subsequently categorized into four groups that aligned with the four questions asked in the focus group interviews: (1) understanding of IPV, (2) cultural influences on IPV, (3) cultural barriers to help-seeking, and (4) practical barriers to help-seeking. Each theme within the categories was enhanced and further detailed through numerous reviews of the transcriptions and field notes. Only themes articulated by three or more participants are included in this paper.

Qualitative research rigor is achieved by focusing on several dimensions: credibility, which assesses the accurate interpretation of data; transferability, which considers the generalizability of the findings to other contexts or populations; dependability, which requires transparency in the research process and researchers; and confirmability, where the outcomes and interpretations must be verifiable by other researchers [21]. To strengthen the rigor of our study, we implemented various strategies. For credibility, we performed member checking with a participant to ensure our findings accurately reflected the participants’ experiences and insights. Regarding transferability, we explored literature on other immigrant communities to compare and contrast experiences. For dependability, we used memoing extensively to record emerging themes and their descriptions, as well as to document our inquiries regarding the data. In terms of confirmability, although researchers inevitably bring their own backgrounds to the study, we meticulously explained our coding decisions and the emergence of themes through the memoing process. This approach ensures transparency in our data coding and analysis, permitting other researchers to validate or question our methodologies and conclusions.

## 3. Results

During the focus groups, participants openly shared their experiences of witnessing or hearing about IPV. Contrary to the researchers’ expectation that the shame culture prevalent in the Korean community might deter them from discussing such sensitive issues, the participants demonstrated a willingness to engage in conversation and discuss cases they were aware of. Four main themes emerged from our analysis of these discussions: (1) differential understanding of IPV, (2) Confucian cultural influences on IPV, (3) cultural barriers for help-seeking, and (4) practical barriers for help-seeking.

### 3.1. Differential Understanding of IPV

All focus groups shared instances of severe domestic violence within their close relationships and how they made sense of those incidents. However, there were important differences in how they understand IPV. Three subthemes emerged from the discussions: (1) comprehensive understanding of IPV (all groups but older male group), (2) female-to-male IPV (all groups but older male group), and (3) intergenerational transmission of violence (both younger female and male groups).

#### 3.1.1. Comprehensive Understanding of IPV

All the groups unanimously recognized physical abuse as a form of violence. However, compared to older male participants, all female and younger male participants demonstrated a clear understanding that violence extends beyond physical harm, including psychological and emotional abuse and exercising control over a partner. Though younger male participants acknowledged verbal, economic, and immigration abuse as a type of violence, female participants showed a more comprehensive and nuanced understanding of IPV. Both female groups identified a wide range of types of violence, including emotional abuse, gaslighting, manipulation, coercion, pressuring, stalking, and sexual abuse. When researchers asked what abuse is, a younger female participant said it’s “kind of like demanding too much, such as trying to take up all of their partner’s time—as if they are entitled to do it to their partner”. Other participants suggested it was “being manipulative. For example, trying to manipulate their partner with their power, maybe with financial power”, and “threatening. Something like ‘if you don’t treat me like this, I am going to leave you.’ … ‘Do you think you can meet anyone better than me?”.

#### 3.1.2. Female-to-Male IPV

The discussion from all groups, except for the older male group, indicated an understanding that violence is not solely male-to-female but can also occur in the opposite direction. The context of immigration was identified as a key element amplifying feelings of isolation in both female groups, leading some immigrant women to be emotionally overdependent on their male partners. This dependency, in turn, led to instances where some women resorted to physical or emotional violence against their husbands. One older female participant remarked:


*In immigrant communities, avenues for women to express their feelings might be limited, making it feel somewhat isolated. When women don’t find an outlet for their emotions… it can potentially lead to intimate partner violence. … there are times when they act out, possibly misdirecting their pent-up emotions towards their male partner. Physical acts like throwing objects or verbal abuse are common, …consistent verbal abuse or physical abuse is no less severe. (Older female participant)*


#### 3.1.3. Intergenerational Transmission of Violence

Both the younger male and female groups discussed intergenerational transmission of violence with examples of dating violence they witnessed. Participants in both groups shared that most perpetrators of dating violence experienced some form of abuse or observed IPV growing up. For example, a young female participant was shocked to learn about the violence in a friends’ relationship, later learning that “both [his] brothers were somewhat like that (using violence to their dating partners), and we found out that their parents were like that too”. Another shared a story of an outwardly devout and gentle man who was privately violent towards his girlfriend. He had been abused by his father growing up. A young male participant talked about a girl who had been physically abused in her childhood and had turned into a perpetrator of violence toward her boyfriend, illustrating how survivors can become perpetrators, regardless of their gender: “Until high school, she got beaten a lot whenever she did something wrong. Then, the girl started dating her boyfriend. And the girl used the same violence to her boyfriend whenever they had a conflict”.

### 3.2. Confucian Cultural Influences on IPV

All four focus groups acknowledged that the main contributing factor to IPV among Korean immigrants is the Confucian belief that men are superior to women (Korean: 남존여비 [nam-jon-yeo-bi]). In terms of cultural factors contributing to IPV, two subthemes emerged from the analysis: (1) strict gender roles and gender inequality (all groups), and (2) cultural tradition of female self-sacrifice (younger and older female groups).

#### 3.2.1. Strict Gender Roles and Gender Inequality

All groups acknowledged a deeply ingrained patriarchal culture, which is heavily influenced by Confucian ideas, perpetuating strict gender roles, gender inequality, and the risk of IPV. Participants in younger male and female groups, in particular, emphasized the strict gender roles associated with traditional Korean culture. One young male participant observed that gender roles and inequality were tied to economic power in older generations, leaving women feeling “quite powerless or helpless”. His sense that this inequality has “diminished somewhat nowadays” was contradicted by a younger female participant who lamented the seemingly “inevitable” nature of gender inequality in Korean immigrant culture:


*Isn’t the relationship between men and women inherently hierarchical? Even if both men and women have the same education and status in society, women are considered lower when they are home.… You see, mothers, mothers-in-law, and older women are working in the kitchen at home. Fathers are sitting in the living room. Seeing this, as a woman myself, it feels odd not to help my mother. Even if I have graduated from a prestigious graduate school, I can’t just hang out there while all the older women are working in the kitchen, so I naturally end up standing up and following them. Then, of course, the men, like our fathers, naturally don’t do it as they are sitting on the couch. It seems like a repetitive cycle. (Younger female participant)*


During the discussion among younger female participants, they pointed out that the persistence and perpetuation of gender norms is not just a problem with men. Since women have observed and learned this custom throughout their lives, they also have a tendency to naturally follow it even though they question it. Another young female participant mentioned that when younger people attempted to deviate from the traditional gender norm, “[Older] people would give subtle remarks discouraging me from doing so”. An additional young female participant agreed by stating that “Korean people still expect traditional gender norms to be followed, even though they do not explicitly state it”.

Women in the older group observed that strict gender roles still persist, and women are expected to conform to the traditional gender roles even with their in-laws. One comment illustrates a cultural lag, where older generations maintain traditional patriarchal Korean values despite Korea’s progress in gender norms:


*Another case is when families immigrate to the United States with their in-laws. While Korea is rapidly changing and adapting [to changing gender norms], in the U.S., the beliefs of the older generation who immigrated long ago still prevail, leading to a lot of conflicts due to the very traditional, patriarchal attitudes of the in-laws. This seems to have a significant impact on the next generation of couples. This unnecessary harassment to daughters-in-law by the in-laws and the stress it causes seems to be a contributing factor to IPV among married couples. (Older female participant)*


This comment from an older women group shows this adherence to outdated norms could be a contributing factor that leads to conflicts, particularly between in-laws and the couple, straining family relationships and affecting marital dynamics.

Regarding this culture, participants of both the younger female and male groups shared feelings of helplessness, noting, “It feels like older man holding traditional values are unchangeable [and] there is nothing we can do about it”.

All groups recognized and acknowledged that patriarchal cultures with Confucian ideas create an environment in which women are more vulnerable and men are often in power and control. Such dynamics can create situations in which IPV is more likely to occur due to the perpetuation of unequal power dynamics and the normalization of women’s subjugation. Participants in the older male group mentioned a case illustrating how the confusion idea of nam-jon-yeo-bi, ‘men are superior to women,’ contributed to IPV: 


*There’s a couple in our church who ha*
*s been married for 43 years. The husband is a very traditional Korean man. He holds onto the belief of male superiority … Recently, he ended up hitting his wife, and now they are in the process of getting a divorce. (Older male participant)*


#### 3.2.2. Cultural Tradition of Female Self-Sacrifice

The strict gender roles and inequality arising from Confucian and collectivist culture contributed to a tradition of female self-sacrifice in Korean immigrant families, which also increases the risk of IPV. This subtheme was identified primarily by female participants. A participant in the older female group observed, “The mother and daughter always have to sacrifice for the family, especially for the son and the father. So, this patriarchal culture persists even after immigration, where women are expected to endure anything for their children and the family”. Another commented, “As you may have read from the novel, Kim Ji-Young: Born 1982, a mother and a daughter should sacrifice for a son and a father”. 

### 3.3. Cultural Barriers to Help-Seeking

Participants also discussed cultural barriers to help-seeking in the case of IPV. Our analysis of this topic revealed three subthemes: (1) shame culture and fear of losing face (all groups), (2) consideration of IPV as a private matter (both younger and older male groups), and (3) the importance of keeping the family intact (all groups but younger female group).

#### 3.3.1. Shame Culture and Fear of Losing Face

Shame and fear of losing face is a significant, persistent, and central theme as a cultural barrier among Korean immigrants to disclosing IPV and seeking help. Shame culture among Korean immigrants is associated with being highly self-conscious of other people’s perceptions of themselves, including being afraid of being judged and ostracized by family members or groups. Confucian culture highly values ‘face’, which is a collectivistic property that represents the entire family [22]. Disclosing family problems and seeking help could mean ‘losing face’ and it is considered as bringing shame to the entire family [23]. All groups agreed that shame culture and fear of losing face play an important role in discouraging survivors from disclosing abuse and seeking help from others.

Participants in the younger groups described a strong cultural emphasis on maintaining a positive public image and managing perceptions, particularly in the context of social media, where an ideal life is often depicted. A participant in the younger male group said, “Korean people seem to have a culture of being overly conscious about how others view them… Koreans tend to overthink and are too conscious of others’ eyes. … Because of this culture, there’s a tendency to keep things hidden”. A younger female group participant concurred, “The problem with our traditional culture is the excessive concern for face and how others perceive us”. She continued, “Koreans care a lot about appearing to be living better than others, doing better than others, and showing off to others. Especially in the context of IPV, it seems that the idea of being humiliated or disclosing pain to others is even more undesirable”.

As a result of this societal norm, experiencing IPV in one’s family is characterized by significant public humiliation, which drives survivors to hide the severity of their suffering to maintain a positive reputation. An older female group participant also highlighted that shame is one of the most significant factors deterring women from reporting IPV, with the family’s reputation and honor taking precedence over addressing the abuse, leading to prolonged silence and inaction.


*In Korean culture, when any kind of violence occurs within the family, there is a reluctance to let it be known outside because of shame. It’s seen as a source of shame and dishonor for the family, so people don’t report it or tell others about it until it becomes really dangerous. The situation gets worse, and the violence intensifies. (Older female participant)*


Additionally, an older male participant highlighted the intense pressures within religious Korean communities, where the exposure of family issues can be particularly humiliating, thereby amplifying the fear of losing face.


*Many Koreans are churchgoers or pastors, and if their family situation (abuse) is exposed, it’s like cutting off your nose to spite your face. They might suffer a lot and know what they should do (reporting it), but they can’t do it … because they are afraid that their family matters will be revealed. There are many people who suffer in silence. They just suffer quietly. (Older male participant)*


Collectively, these insights reveal that public image and family honor play a significant role in Korean culture, leading to a tendency to endure IPV in silence, often until it becomes severe. Focus group participants highlighted the critical role of shame culture and fear of losing face in perpetuating IPV within the Korean immigrant community.

#### 3.3.2. Consideration of IPV as a Private Matter

Men in both the younger and older groups tended to perceive IPV as a private matter that should be resolved within the couple and that outsiders should not intervene in the couple’s conflict. However, the reasons for this tendency were identified differently among male groups. Older participants discussed IPV as well as child abuse and asserted why they perceive family violence as a private matter and why no one should intervene. The first reason is shown in the Korean traditional saying ‘누워서 침 뱉기’ meaning ‘Cutting off one’s nose to spite one’s face,’ which refers to an unnecessary overreaction that is not helpful to resolve the problem but self-destructive. A member of the older male group said that people cannot report IPV because they think, “It is literally cutting off your nose to spite your face!” The next reason is a belief that conflicts between couples are temporary and common and will eventually be resolved by themselves. An older man shared, “Marital fights are like cutting water with a knife (Korean: 칼로 물 베기), so in Korea, the police don’t want to get involved. They just let them fight”. ‘Cutting water with a knife’ is an old Korean saying that a quarrel between a husband and wife is easily reconciled, as described by another older male:


*You can intervene in everything else, but you should never interfere in a couple’s conflicts because, you know, they might be at each other’s throats one moment and then make up sooner or later. … That’s why I’ve learned that you can intervene in anything else, but you should never get involved in a couple’s fight. (Older male participant)*


This perception is likely to lead to the assumption that all couple quarrels are common and do not require intervention, which may result in the minimization of the severity and impact of violence within a couple’s relationship. Moreover, referring to children reporting their parents, another expressed, “American law is wrong. I have a lot of negative opinions about American laws”. The older male group shared that disclosing IPV is not beneficial but self-destructive and damaging to themselves and their family’s reputation.

Statements from younger males differed from older males in their reasons for IPV remaining a private matter. A comment by one younger male suggests a belief that intervention by outsiders will only be temporary and that the couple will ultimately have to resolve their issues on their own. “Even if an outsider intervenes to solve the problem or not, at the end of the day, that person will leave the relationship… So, ultimately, the responsibility to resolve any remaining issues or conflicts lies with themselves. That’s probably why people don’t ask for help”.

#### 3.3.3. Importance of Keeping the Family Intact

Except for the younger female group, all other focus group participants highlighted the importance of keeping the family intact and putting the family’s interests first as important cultural barriers to seeking help. The younger male group articulated a cultural expectation that families should remain together in order to protect the children, which perpetuates the mindset of enduring violence until the children become independent.


*In Korean culture, compared to Americans, parents tend to think that they need to protect their kids in a way that stay together no matter what happens. Because of this idea, people have this mindset that I need to tolerate until their kids grow up and become independent. Children also tend to think that I have to tolerate it because they are their parents. (Younger male group participant)*


Furthermore, the fear of family breakdown is an important deterrent to addressing IPV, as it is viewed as worse than the continuation of violence. In this perspective, familial unity is valued over individual safety and well-being within a broader cultural context. According to the older female group, women are expected to endure hardships for their children and families. One woman stated, “There is a strong cultural belief that women must endure anything for the sake of the children and the entire family, no matter what happens”. This expectation also normalizes their suffering as a necessary part of the family’s maintenance.

Older male participants emphasized a sense of shame associated with the disclosure of IPV to the public, suggesting a broader societal tendency to prioritize maintaining the appearance of a functional family over addressing the underlying issues of violence and abuse. One participant’s account of his friend’s request to convince the wife to give up divorce proceedings and the questioning of her motives after enduring a long marriage reflects this tendency.


*He choked his sleeping wife in the middle of the night, causing the police to come. … And now he keeps coming to me, asking me to talk to his wife and tell her he won’t do it again. But when I called her, she was very determined, saying she feels wronged for having lived like this for 43 years. So, in my opinion, they are both the same kind of person. They’ve lived together for 43 years because they are the same. Then, they should have gotten divorced earlier, why do it now? Just shake it off and get back together since he apologized. That’s what I was trying to say. It’s better than getting a divorce. If you get divorced, then what? Are you going to find another man? You have children, so isn’t it better to be a bit more thoughtful? (Older male participant)*


Participant data demonstrate a belief in the importance of keeping families together. This, combined with gender-specific expectations and stigmas associated with family breakdowns, creates significant barriers to seeking help for IPV within the Korean community. Family unity and the shame associated with divorce or separation discourage survivors from seeking help, often trapping them in abusive situations.

### 3.4. Practical Barriers to Help-Seeking

In terms of practical barriers to help-seeking, participant data underscore Korean immigrants’ unique challenges and complexities of navigating support systems in a foreign country. We identified five practical barriers to seeking help for IPV: (1) Language barrier (all groups), (2) Financial ramifications (both younger and older female groups), (3) Lack of resources and accessibility (all but older male group), (4) Problems related to immigration status (both younger groups), and (5) Lack of social support (older female and male groups).

#### 3.4.1. Language Barrier

In every focus group discussion, language barriers emerged as a major obstacle to seeking help for IPV. Younger and older female group participants, in particular, discussed this barrier more in-depth because disclosing IPV entails not only asking for help but also facing the fear of navigating life alone in a foreign country, which can be scary for women who are not fluent in English. A participant in the younger female group explained,


*Coming here as an immigrant, the language barrier seems to be a significant issue. Conversations are not as comfortable as in Korean, and when something like this happens, (they would constantly think) ‘what should I say?’ You need to know English to a certain extent, to be able to handle such situations. But that can be intimidating. There could be misunderstandings if it’s not spoken in a right way. So, people might rely more on Korean-speaking individuals like pastors at church. It seems harder for them to approach professionals directly. (Younger female participant)*


For some Korean women immigrants, it is not only difficult to communicate in English, which affects daily necessities such as getting a job, but it significantly impedes one’s ability to seek help or contemplate leaving an abusive relationship. An older female participant explained, “Because of the language barrier, even if they consider living separately, fear of navigating everything alone might lead them to endure and continue to live in the abusive situation”.

#### 3.4.2. Financial Ramifications

Participants also discussed financial ramifications as one of the primary practical barriers to help-seeking. Similar to the language barrier discussion, both younger and older female participants highlighted this challenge. A participant in the older female focus group explained this barrier:


*The reason is that, while there are women who have a job, there are also those who don’t. In cases where the husband is the sole earner, if the wife reports violence, she can no longer receive financial support from him, which is why I think economic issues are the biggest problem. That’s why they can’t report it, and it becomes a vicious cycle. (Older female participant)*


A young female participant expressed a similar idea, “Financial aspects also seem important. If someone has been a survivor of violence… and sought help, then they have to take full responsibility for their own life. That means they need to protect themselves”. She explained, “The most important tool for self-protection is having financial power”. The data from the younger and older female groups suggest there is a critical link between economic dependency and help-seeking behavior for IPV among Korean immigrant women. Women’s financial dependence on their abusers significantly limits their ability to leave abusive situations. Because of this economic dependency, reporting abuse or seeking help is difficult, keeping them in a cycle of violence.

#### 3.4.3. Lack of Resources and Accessibility

All groups, except for the older male group, identified a lack of knowledge about IPV and accessibility to resources as major barriers to seeking help, often due to language barriers and isolation. A younger female participant highlighted the lack of exposure to information and resources, noting she was unaware of any IPV social services in Atlanta despite living there for over a decade. “Despite living here for over a decade, our ignorance suggests we haven’t been sufficiently exposed to or made aware of them”. This reflects a broader community-level lack of awareness, preventing effective service utilization. She suggested disseminating information in community hubs such as local restaurants and churches to build awareness among immigrants. A younger male participant emphasized the structural differences between Korea and the U.S., pointing out Korea’s established hotlines compared to the U.S.’s general 911 system. Even though social services exist for IPV intervention in the U.S., there is a notable lack of awareness and accessibility among Korean immigrants. An older female participant also mentioned, “We often don’t know about these resources or how to use them”.

#### 3.4.4. Problems Related to Immigration Status

Immigration status was identified as a significant factor in the decision to seek help for IPV by both the younger male and female groups and the older female group. They identified the fear of deportation and/or potential separation from children as a major deterrent to reporting IPV to authorities. This concern was highlighted during a discussion in which a younger female participant mentioned that visa status often prevents survivors from reporting abuse. “One possible reason (for not seeking help) is their visa status”, with many participants nodding in agreement at this observation and expressing further concern about the risk of deportation, especially for those who are undocumented. Another younger female participant added, “Reporting the abuse could lead to deportation if they are undocumented”.

#### 3.4.5. Lack of Social Support

Both older female and male participants identified that a lack of social support is one of the significant barriers to seeking help. Participants emphasized the lack of close, trusted relationships in the immigrant community. An old male participant noted that, back in Korea, they used to experience strong social networks, including so-called ‘testicle friends’ (literal translation of the Korean word for male friendship: 불알친구 [Bull-Ahl-Chin-Goo]) (i.e., lifelong male friendship with deep trust, similar to ‘bosom buddies’ or ‘best friend’ in English), and this helped resolve many relational issues in the family. One of the older male participants said:


*Neighborhoods are your closest friends, and Korean proverbs say you have ‘testicle friends’. They’re friends who grew up together, lived in the same neighborhood, and their fathers and grandfathers are friends, so there’s no deceit between them. … Since there is no deceit, everything can be resolved. … But in this immigrant society, there are no friends, no neighbors like that. (Older male participant)*


Likewise, an older female participant also mentioned that “in Korea, you can go to another family member’s or parent’s house to seek refuge or ask for help. However, here, there’s nowhere to escape to and no one to talk to”. They highlighted the importance of having close friends and extended family support systems that provide a safety net and facilitate conflict resolution. The lack of such support in the immigrant context leaves individuals isolated and without trusted confidants to turn to in times of need.

## 4. Discussion

The focus groups aimed to explore how different genders and ages perceive IPV among Korean immigrants and the barriers they face when seeking help. We observed that the differences between genders and age groups varied, with the most significant disparities appearing between older men and other groups. Younger groups and the older female group showed feelings of powerlessness against the power held by older Korean men. This disparity is largely due to the older males being deeply influenced by patriarchal and Confucian values, holding significant cultural authority in Korean immigrant society. Confucianism promotes a clear hierarchy where older men wield substantial influence [2,24]. In this regard, Tung [23] contends that Asian Americans who are less acculturated are more inclined to adhere to traditional Confucian doctrines, which emphasize collectivist values and distinct and strict gender roles. Given that 80% of participants from the older male group primarily speak only Korean, it is reasonable to assume that this segment of male immigrants is more likely to align with these traditional values.

### 4.1. Differential Understanding of IPV

Our analysis revealed a differential understanding of IPV between groups. Compared to the older male group, the other groups discussed different types of violence in greater depth. This result is consistent with Delgado Álvarez et al. [25], which highlighted gender and cultural effects on perceptions of psychological violence. In their study, women displayed greater awareness and perception of psychological violence than their male counterparts. This suggests that women are generally more attuned to the subtleties and complexities of IPV, including non-physical forms such as coercion, manipulation, stalking, etc.

Considering that only women and younger males in our study were able to discuss IPV in greater depth beyond physical violence, increasing knowledge of IPV and targeting older men is critical to potentially reducing IPV within the Korean immigrant community. Taking into account that Korean immigrants and their descendants are more likely to rely on church communities for family issues than community services [26,27], a potentially effective method is targeting Korean American pastors, who hold influential positions and can play a pivotal role in educating and shifting perspectives within this demographic [28,29].

Unlike older groups, younger groups discussed the intergenerational transmission of violence with real stories that they had witnessed. Several researchers have suggested potential mechanisms with behavioral genetics, the impact of perinatal exposure to trauma, a consideration of the neurobiological mechanisms of transmission, and the role of developmental psychopathology (social learning and attachment) in the transmission process [30,31]. This suggests that initiatives and interventions must go beyond addressing immediate crises. Indeed, such narratives indicate that addressing IPV requires an integrated approach that encompasses not only providing assistance to the immediate survivors but also long-term support and education for families, particularly children, that are exposed to the abusive environment.

### 4.2. Confucian Cultural Influences on IPV

The acknowledgment across all four focus groups that Confucian beliefs, particularly the principle of male superiority (남존여비 [nam-jon-yeo-bi]), significantly contribute to IPV underscores the pervasive impact of cultural norms on interpersonal dynamics. This broad recognition of strict gender roles as a factor in IPV reflects entrenched cultural norms that not only perpetuate male dominance but also legitimize control and violence as mechanisms of maintaining that dominance. This Confucian culture not only increases the risk of IPV but also impacts help-seeking behavior. Cuesta-García & Crespo [1] illustrate how rigid gender norms affect immigrant survivors’ help-seeking behavior as well. These roles impose a strong sense of duty to maintain family unity and normalize abusive relationships, which discourages women from seeking help. These dynamics highlight the need for IPV interventions to address deeply rooted gender expectations and provide support that empowers family and community members to overcome cultural barriers to seeking help.

Moreover, the cultural tradition of female self-sacrifice, particularly noted by female groups, highlights how cultural expectations can trap women in abusive relationships. A culture that expects mothers and daughters to sacrifice their own well-being and needs for their male family members is more likely to perpetuate a culture of women’s subordination to men and can contribute to the acceptance or normalization of abuse, particularly if it is seen as beneficial to the family [32,33]. The pressure on women to endure hardships for the sake of the family reinforces gender inequalities and can exacerbate women’s vulnerability and risk of IPV. Lee & Hadeed [34] indicate that such cultural norms not only minimize women’s experiences of violence but also socially and psychologically compel them to endure abuse for the ‘greater good’ of the family unit, often at great personal cost.

Interventions should go beyond providing legal aid and escape options by actively challenging the cultural narratives that contribute to IPV. In order to effectively challenge and transform such narratives, educational campaigns and community-based programs are crucial to openly addressing the detrimental effects of cultural expectations and providing robust support to survivors. This can significantly shift public perception and lessen the stigma around leaving abusive situations. As discussed earlier, involving older men in these educational efforts is critical to addressing gender inequality and the normalization of violence, helping to reshape perceptions of couples’ relationships that are based on mutual respect and equality. A comprehensive approach that integrates legal, educational, and social strategies is necessary to counteract the cultural norm of female self-sacrifice. Such an approach would empower survivors to seek help without fear of societal judgment, effectively breaking the cycle of violence and enhancing the overall well-being of IPV survivors in the community.

### 4.3. Cultural Barriers to Help-Seeking

Cultural barriers to help-seeking for IPV among different age groups provide critical insight into cultural dynamics affecting these behaviors. The three subthemes identified in the analysis each speak to deeply ingrained cultural norms that can significantly hinder the willingness and ability of individuals to seek help for IPV. The shame culture and fear of losing face is a pervasive issue across all demographic groups and is universal in Asian countries that share Confucian culture [35,36,37]. Fear of losing face can prevent victims from seeking help or even acknowledging abuse publicly due to the potential social repercussions that could reflect negatively on them and their families [23]. This barrier is deeply rooted in cultural expectations about maintaining personal and familial honor in public settings [23,26,38].

Shame culture and the fear of losing face are deeply entwined with the idea of IPV being a private matter in a way that both contribute to keeping IPV hidden [39]. The belief that IPV should be a private issue is particularly noted among male groups, as disclosing IPV would be self-destructive and damaging to their reputation. Due to the importance of maintaining a positive reputation as a family, fear of judgment and shame can lead to reluctance to report [39,40]. Korean immigrants believe utilizing community services can lead to public embarrassment and loss of face for both the individual and the family, which is believed to be detrimental to the whole family [41]. Immigrants from other Asian countries have also shown that protecting their faces, reputations, and family harmony prevents them from seeking help [42,43]. This perspective can lead to the underreporting of IPV incidents and a lack of external intervention, as it is seen as an internal family or marital issue that should be resolved privately. Such a viewpoint not only minimizes the severity of IPV but also isolates the victims, limiting their access to external support and resources and prolonging the abuse.

The theme of the importance of keeping the family intact reflects a common cultural expectation that values the cohesion, stability, and harmony of the family unit over the individual well-being of its members. This emphasis can pressure victims to endure abuse rather than disrupt the family structure [34,41,43]. Interestingly, the younger female group did not mention this aspect at all, possibly indicating a generational shift in attitudes regarding family obligations and individual rights.

The focus group discussion shows cultural barriers to help-seeking for IPV underline the complex interaction between deeply ingrained cultural norms and immigrant survivors’ behaviors. Understanding these barriers is crucial for developing effective interventions and support systems tailored to specific cultural contexts. Considering the pervasive influence of shame culture and the fear of losing face, any efforts to encourage help-seeking behaviors must address the stigma associated with IPV [44]. Public awareness campaigns and community-based interventions should aim to shift cultural perceptions, emphasizing that seeking help is a sign of strength and responsibility rather than a source of shame or a sign of weakness. These efforts could include educating the community about the harmful effects of IPV and the importance of reaching out to support systems, thereby reducing the stigma and social repercussions of reporting abuse [44].

The belief that IPV is a private issue, particularly among male groups, shows that they are more sensitive to keeping face and family reputation, which highlights the need for targeted interventions that address the unique challenges faced by men in these cultural contexts. Programs that promote healthy masculinity and provide safe spaces for men to discuss IPV without fear of judgment could be beneficial [45]. Additionally, involving respected community leaders such as pastors and influencers in these efforts could help change the narrative around IPV and encourage more men to seek help navigating healthy and positive communications [29,46].

In light of the emphasis on keeping the family intact over individual well-being, interventions can also focus on redefining what it means to have a harmonious and stable family. Education and counseling services can help immigrants understand that a healthy family environment is one free from violence and that seeking help is essential for the well-being of all family members. Support services that offer confidential and culturally sensitive assistance can help survivors feel safer in coming forward.

### 4.4. Practical Barriers to Help-Seeking

Practical barriers compounded by the immigration context further exacerbate the difficulties faced by those seeking help. The five practical barriers identified by our participants highlight the unique and complex challenges Korean immigrants face in a foreign country. Like immigrants from other countries [1,2,14,43], language barriers make it difficult for Korean immigrants to communicate their needs and understand the support available to them. Financial dependence is also a significant obstacle for survivors, preventing them from seeking help. In terms of lack of resources and accessibility, the scarcity of tailored support services for Korean immigrants indicates systemic gaps in assistance and is a common barrier immigrants face [28]. Immigrants may also have concerns about the potential for deportation or other vulnerabilities with regard to their residency and legal rights [14,26,47]. Lastly, both older groups mentioned that isolation and limited social networks hinder help-seeking. This suggests that older individuals are more vulnerable to isolation and have a greater need for social support [41].

These findings imply that addressing practical barriers to help-seeking requires a multi-faceted approach tailored to the specific needs of Korean immigrants. First, providing multilingual resources and employing bilingual staff in IPV support services can bridge the communication gap. Second, financial assistance and economic empowerment programs are essential for victims to achieve economic independence from their abusive partner. Third, it is imperative to enhance resources and accessibility. Establishing more IPV support centers within immigrant communities and ensuring these services are well-publicized and easily accessible is crucial. Fourth, it is essential to inform immigrants about immigration protections, such as the Violence Against Women Act and U visa. Awareness of legal reforms and protections that enable IPV survivors to seek help without fear of deportation or legal repercussions can encourage more survivors to come forward. Lastly, building and strengthening social support networks will decrease isolation among immigrants, considering isolation is a risk factor for IPV and help-seeking. Community programs that foster connections and provide emotional support can reduce isolation among older immigrants, making it easier for them to disclose and seek help. By addressing these barriers, support systems can become more inclusive and effective.

### 4.5. Strengths and Limitations

The focus group study’s strengths include its ability to provide in-depth insights into Korean immigrants’ perceptions of IPV and barriers to seeking help and how these differ by gender and age. The diverse participant groups and identification of specific cultural and practical barriers enhance the comprehensiveness of the findings and provide actionable insights for policy and practice. However, there are several limitations to consider. Since there was no specific requirement for participants to have personal experience with IPV, our study might have limited insights into actual survivors’ lived experiences and help-seeking behaviors. Additionally, the study has limited generalizability due to its focus on a group living only in a metropolitan city in the Southeastern United States, a small sample size, and selection bias due to the snowball sampling process. Participants’ self-reported answers may be biased due to social desirability. This bias is likely to be further reinforced by the face culture prevalent in the Korean community and religiosity, considering that younger groups were recruited in the Christian church. Nevertheless, we found that many participants were quite candid in their discussion of IPV. Future research should aim to include a broader range of participants from different immigrant communities in various areas, such as other metropolitan areas as well as rural areas. Also, researchers could use a mixed-methods approach to gather quantitative data and conduct longitudinal studies to examine the long-term impact of barriers.

## 5. Conclusions

This study sheds light on Korean immigrants’ perceptions of IPV and cultural and practical barriers to seeking help, with an emphasis on age and gender differences. The findings highlight significant disparities, most notably between older men and other demographic groups. Practical barriers such as language barriers, financial constraints, lack of accessible resources, immigration-related fears, and social isolation are critical obstacles that hinder help-seeking behavior. These challenges are deeply intertwined with cultural norms, such as the shame culture and the emphasis on family reputation, which further complicate the ability of victims to seek assistance. However, Confucian culture also has positive aspects, such as a focus on group-focused familial values that facilitate collaboration, a value placed on harmony within the family that facilitates mutual support, and a respect for parents and elders that promotes intergenerational connections. Recognizing these aspects can inform culturally sensitive interventions that leverage these strengths to effectively change community perceptions toward IPV and promote healthy relationships.

By identifying community perceptions of IPV and barriers to help-seeking, the study underscores the need for culturally sensitive and age- and gender-specific interventions that address the unique needs of immigrant communities. Moving forward, it is essential to develop targeted support systems, enhance legal protections, and foster community awareness to create a more supportive environment for IPV victims. The study provides a foundation for future research and intervention strategies aimed at changing community perceptions and attitudes toward IPV to improve the help-seeking behaviors and overall well-being of Korean immigrants and other vulnerable populations.

## Figures and Tables

**Table 1 ijerph-21-01508-t001:** Sociodemographic characteristics of focus group participants (n = 38).

	Younger Females (n = 10)	Younger Males (n = 10)	Older Females (n = 8)	Older Males (n = 10)
Characteristic	n (%)	n (%)	n (%)	n (%)
Age				
Range	23–35	22–34	50–63	73–81
Mean	29.1	29	56	76.6
Year of residency (mean)	15.3	15.56	29.43	34.4
Birthplace				
U.S.	1 (10%)	1 (10%)	0 (0%)	0 (0%)
Korea	9 (90%)	9 (90%)	8 (100%)	10 (100%)
Religious group				
Catholic church	0 (0%)	0 (0%)	4 (50%)	1 (10%)
Protestant church	10 (100%)	8 (80%)	3 (37.5%)	8 (80%)
Buddhist temple	0 (0%)	0 (0%)	0 (0%)	0 (0%)
None	0 (0%)	2 (20%)	1 (12.5%)	1 (10%)
Education				
Less than high school	0 (0%)	0 (0%)	0 (0%)	4 (40%)
High school	2 (20%)	2 (20%)	0 (0%)	4 (40%)
Some college	0 (0%)	4 (40%)	0 (0%)	0 (0%)
College	4 (40%)	3 (30%)	4 (50%)	1 (10%)
Graduate school	4 (40%)	1 (10%)	4 (50%)	1 (10%)
Primary language				
Korean	3 (30%)	3 (30%)	5 (62.5%)	8 (80%)
English	0 (0%)	0 (0%)	0 (0%)	0 (0%)
Both	7 (70%)	7 (70%)	3 (37.5%)	1 (10%)

## Data Availability

The data in this study is not available, as participants did not consent for the data to be shared with others other than the researchers of the study.

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
