# Peer review of "Cultural and Practical Barriers to Seeking Help for Intimate Partner Violence Among Korean Immigrants: Exploring Gender and Age Differences"

_ijerph, 2024, doi:10.3390/ijerph21111508_

Round 1
Reviewer 1 Report
Comments and Suggestions for Authors
While the study offers valuable, though well-known, insights into the cultural and practical barriers Korean immigrants face when seeking help for intimate partner violence (IPV), I believe there is a notable limitation in the research design that warrants attention.
The inclusion criteria for the study participants, which required them to self-identify as Korean or Korean American, speak Korean, be 18 years or older, and live in Georgia, may not fully align with the study's objectives. Given the study’s aim to explore perceptions and barriers to help-seeking for IPV, I believe it would have been important to include participants who had direct experiences of IPV while living in the United States. This criterion would ensure that the focus group discussions reflect the lived experiences of IPV survivors within the U.S. context. However, the current criteria allowed for participants who may not have directly experienced IPV, as highlighted by one of the questions asked of participants, which referred to "issues you’ve seen or heard within the Korean community regarding family violence."
While this may still yield important community-level perspectives, it limits the depth of insight into the help-seeking behaviors of actual IPV survivors. I noted that this limitation was not addressed in the "Limitations" section of the article.
I hope this feedback is helpful and contributes to a deeper understanding and future refinement of qualitative research on this topic. Thank you for your consideration of this input.
Author Response
Comments 1: While the study offers valuable, though well-known, insights into the cultural and practical barriers Korean immigrants face when seeking help for intimate partner violence (IPV), I believe there is a notable limitation in the research design that warrants attention.
The inclusion criteria for the study participants, which required them to self-identify as Korean or Korean American, speak Korean, be 18 years or older, and live in Georgia, may not fully align with the study's objectives. Given the study’s aim to explore perceptions and barriers to help-seeking for IPV, I believe it would have been important to include participants who had direct experiences of IPV while living in the United States. This criterion would ensure that the focus group discussions reflect the lived experiences of IPV survivors within the U.S. context. However, the current criteria allowed for participants who may not have directly experienced IPV, as highlighted by one of the questions asked of participants, which referred to "issues you’ve seen or heard within the Korean community regarding family violence."
While this may still yield important community-level perspectives, it limits the depth of insight into the help-seeking behaviors of actual IPV survivors. I noted that this limitation was not addressed in the "Limitations" section of the article.
I hope this feedback is helpful and contributes to a deeper understanding and future refinement of qualitative research on this topic. Thank you for your consideration of this input.
Response 1: Thank you for your feedback.
While there have been numerous studies focusing on the experiences and barriers faced by IPV survivors among immigrants, the primary aim of our study was to explore cultural factors within the Korean community that could act as barriers to seeking help. Our intention was not to focus exclusively on IPV survivors but to understand broader community perceptions that could inform the development of community-wide interventions that target changes in attitudes toward IPV and help-seeking. Therefore, we opted for focus groups that included community members rather than restricting our study solely to survivors. This approach allowed us to gather a wide range of perspectives on the cultural aspects that may influence attitudes toward IPV and help-seeking behaviors among Korean immigrants.
We made a change in limitation. This change can be found at page 15 and line 711-714 “However, there are several limitations to consider. Since there was no specific requirement for participants to have personal experience with IPV, our study might have limited insights into actual survivors’ lived experiences and help-seeking behaviors. Additionally, the study has limited…”
Also, we clarified the purpose of the study to reduce confusion for readers. This change can be found at page 2, line 84-90. “The purpose of our study was to explore Korean immigrants’ community members’ perceptions of IPV and cultural and practical barriers to seeking help, with a focus on age and gender differences. We employed a qualitative approach to gain a deep understanding of the perspectives and insights of this target population, aiming to inform culturally responsive interventions, targeting the entire Korean immigrant community to effectively change perceptions and attitudes toward IPV.”
Reviewer 2 Report
Comments and Suggestions for Authors
Thank you for the opportunity to review this paper. It is an extremely important work and has useful practical recommendations. It was also very interesting to read.
It would be good to include a little more detail about ethics e.g., ethics approval number and how researchers protected participants. Given the sensitive topic this is important in itself, but can also provide a model to those who are doing similar work.
Some of the references for incidence or prevalence seemed to be secondary citations. If there are any primary sources, please include them.
I have an idea why you didn’t recruit people with lived experience of IPV or ask about their direct experience of IPV. Apart from not identifying anyone in the focus groups, do Koreans tend to prefer indirect questioning to direct questioning? I think it would benefit your audience to clearly justify this aspect of the research, because some readers might question why you didn’t interview people with lived experience.
In terms of the transcription from Korean to English, were there any mechanisms to ensure accuracy, such as back translation by another author? If not, why?
A little more detail about the comparative analysis of codes would be useful in terms of transparency, because this term is sometimes used in different ways.
Did this work draw on any particular theoretical frameworks in terms of IPV? It would be good to include this or indicate why not.
The work was well structured in terms of results. The headings and subheadings made this clear.
What is your definition of intimate partner violence? In the data it sounded like they were talking about common couple violence at times and other times about intimate partner terrorism or coercive control. If you left it open so participants could demonstrate their understanding, then it could be useful to mention that, It would also be beneficial to clarify your own position on that, especially because definitions and typologies can be a contested in this area.
I am aware there is a cap on U visas in the United States. Does this have any impact? You don’t need to respond to this if it is not relevant. I am just curious. It seems ridiculous to cap this kind of visa.
One of the strengths of this research is in asking about what is happening in the community rather than direct questions. Although you do not get lived experience, it is a good way to understand community perceptions and cultural influences without causing more trauma or potentially putting participants in danger.
You included good proposals for future research.
It could be good to note some positive aspects of culture. Culture is often blamed for poor health and other outcomes, when culture often has some health promoting or positive aspects as well. Are there any aspects of culture that are promoting positive versions or traits of masculinity either in the data or the literature? Considering these could help to adopt a more strengths-based approach that promotes a more positive masculinity that is still in alignment with culture. Or for those with high religiosity there are aspects of many religions that promote more positive masculinity that are more gender equitable, which could be useful (unfortunately these have often been eclipsed or muted by those who wish to maintain a more patriarchal structure).
It sounded a little bit repetitive at times, particularly in the discussion. I hear some people are moving towards presenting the results and discussion together to prevent repetition. That is not a directive, just something to consider. People and journals have different preferences.
All things considered, I think this is a very good paper and I commend you on your work.
Author Response
Comments 1: It would be good to include a little more detail about ethics e.g., ethics approval number and how researchers protected participants. Given the sensitive topic this is important in itself, but can also provide a model to those who are doing similar work.
Response 1: Thank you for your valuable feedback. We agree with your comment. We rearranged sentences in the “2.1 setting and design section” to add detailed information regarding the ethical considerations of our study. This includes the name of the institution that granted ethics approval, the approval number, and a description of how we protected participant confidentiality. This change can be found on page 3, line111-112.
Specifically, participants were instructed not to reveal any identifiable information regarding the cases they discussed. They were also encouraged to speak only to the extent they felt comfortable, ensuring they could express their views without concerns about privacy or exposure. This additional detail not only addresses the ethical aspects of our research but also serves as a model for others conducting similar studies. This change can be found on page 3, line 139-146.
Comments 2: Some of the references for incidence or prevalence seemed to be secondary citations. If there are any primary sources, please include them.
Response 2: Thank you for pointing that out. We agree with your comment. We included primary sources for the page 1, citation no.3-7, line 40.
Comments 3: I have an idea why you didn’t recruit people with lived experience of IPV or ask about their direct experience of IPV. Apart from not identifying anyone in the focus groups, do Koreans tend to prefer indirect questioning to direct questioning? I think it would benefit your audience to clearly justify this aspect of the research, because some readers might question why you didn’t interview people with lived experience.
Response 3: Thank you for your insightful feedback. The decision not to directly recruit individuals with lived experience of IPV or to ask directly about such experiences was indeed deliberate and rooted in cultural considerations. In Korean communities, both in the U.S. and globally, there is a strong cultural emphasis on privacy and maintaining face, particularly regarding sensitive personal matters such as IPV. Direct questioning about such experiences can be seen as confrontational or disrespectful, potentially leading to discomfort or unwillingness to participate, especially in focus groups where others are present.
Instead, we employed indirect questioning to provide participants with a more comfortable space to discuss IPV within their community context without the pressure of revealing personal experiences. This approach not only respects the cultural norms of discretion and indirect communication prevalent among Koreans but also aligns with our study's broader focus on community and cultural perceptions rather than individual trauma.
In addition, the goal of the study was to explore perceptions and attitudes toward IPV and ultimately develop strategies to effectively intervene at the community level to address this issue. This focus on community intervention aligns with the indirect approach to questioning, enabling us to gather information that can inform broader, culturally sensitive prevention and support programs tailored specifically for the Korean immigrant community. This change can be found on page 2, line 85-96.
Comments 4: In terms of the transcription from Korean to English, were there any mechanisms to ensure accuracy, such as back translation by another author? If not, why?
Response 4: Thank you for pointing this out. Two native Korean-speaking authors were responsible for translating the focus group discussions from Korean to English. After independently translating the transcripts, they compared their versions to resolve any discrepancies and ensure the accuracy of the final English translation. This collaborative approach allowed for a thorough review and refinement of the translated material, enhancing the fidelity of the transcription process. This change can be found on page 5, line 168-171: “Two native Korean-speaking authors were responsible for translating the focus group discussions into English. Both authors independently translated the transcripts and then compared their versions in order to resolve any discrepancies and ensure accuracy.”
Comments 5: A little more detail about the comparative analysis of codes would be useful in terms of transparency, because this term is sometimes used in different ways.
Response 5: Thank you for your feedback. We agree with you. We clarified the process we used in our analysis on page 5, line 174-177. This process entailed comparing codes within and across participants from the different focus groups and organizing the codes according to their similarities and differences.
Comments 6: Did this work draw on any particular theoretical frameworks in terms of IPV? It would be good to include this or indicate why not.
Response 6: Thank you for your feedback. We agree that it would be better to include the theoretical frameworks. Therefore, we made changes in our 2.1 settings and design section. The change can be found at the page 3, line 99-109:
“To understand the complex dynamics of intimate partner violence (IPV) among Korean immigrants, we drew on intersectionality theory as a theoretical framework. In this approach, various forms of systems, such as gender, race, class, education level, language proficiency, cultural heritage, acculturation and immigration status are not separated from each other, but interact and affect individuals in multiple ways (Crenshaw, 2013). An understanding of intersectionality is particularly important when analyzing immigrants' perceptions about how culture and immigration context contribute to IPV and help-seeking behavior. The purpose of our design was to capture the nuanced experiences of IPV that are influenced by the intersection of gender, age, and culture within the Korean immigrant community. We incorporated qualitative methods that allowed participants to discuss their IPV related experiences and perceptions within a culturally contextual framework. “
Comments 7: What is your definition of intimate partner violence? In the data it sounded like they were talking about common couple violence at times and other times about intimate partner terrorism or coercive control. If you left it open so participants could demonstrate their understanding, then it could be useful to mention that, It would also be beneficial to clarify your own position on that, especially because definitions and typologies can be a contested in this area.
Response 7: Thank you for your feedback. I appreciate you pointing that out. When we facilitated focus group discussion, we provided a definition of IPV, including physical abuse, emotional abuse, sexual abuse, and economic abuse with examples. However, during the discussion, participants came up with many other forms of abuse, including manipulation, coercion, stalking, and immigration abuse. We made the change in 2.3. Measures by adding a definition of IPV. The change can be found page 4, line 150-153: “The facilitators introduced the purpose of the study and the procedures by providing a definition of IPV. Facilitators provided a definition of IPV as physical abuse, emotional abuse, sexual abuse, and economic abuse with examples. After explaining the definition, we asked the following overarching questions:”
Comments 8: I am aware there is a cap on U visas in the United States. Does this have any impact? You don’t need to respond to this if it is not relevant. I am just curious. It seems ridiculous to cap this kind of visa.
Response 8: Thank you for asking a very important question. The cap on U visas, indeed, presents a significant barrier for survivors seeking help and legal recourse in the United States. Each year, only 10,000 U visas are available, leading to extensive backlogs and prolonged wait times for applicants. This limitation can deter survivors from coming forward to report IPV due to the uncertainty and lengthy process associated with obtaining legal status through a U visa. As such, this cap not only impacts the accessibility of protections intended for survivors but also underscores the need for policy considerations to address these limitations and improve access to resources and support for IPV survivors within the immigrant community. This aspect is crucial to understanding the broader systemic barriers that affect immigrant survivors’ decisions to seek help.
Comments 9: One of the strengths of this research is in asking about what is happening in the community rather than direct questions. Although you do not get lived experience, it is a good way to understand community perceptions and cultural influences without causing more trauma or potentially putting participants in danger.
Response 9: Thank you for acknowledging the importance of our study.
Comments 10: You included good proposals for future research.
Response 10: Thank you so much.
Comments 11: It could be good to note some positive aspects of culture. Culture is often blamed for poor health and other outcomes, when culture often has some health promoting or positive aspects as well. Are there any aspects of culture that are promoting positive versions or traits of masculinity either in the data or the literature? Considering these could help to adopt a more strengths-based approach that promotes a more positive masculinity that is still in alignment with culture. Or for those with high religiosity there are aspects of many religions that promote more positive masculinity that are more gender equitable, which could be useful (unfortunately these have often been eclipsed or muted by those who wish to maintain a more patriarchal structure).
Response 11: Thank you for your insightful feedback. We appreciate your perspective on this matter. We agree with you. Therefore, we made changes in our conclusion to address positive aspects of the culture as well. The change can be found at page 15-16. Line 734-739: “However, Confucian culture also has positive aspects such as a focus on group-focused familial values that facilitate collaboration, a value placed on harmony within the family that facilitates mutual support, and a respect for parents and elders that promotes intergenerational connections. Recognizing these aspects can inform culturally sensitive interventions that leverage these strengths to effectively change community perceptions toward IPV and promote healthy relationships.”
Comments 12: It sounded a little bit repetitive at times, particularly in the discussion. I hear some people are moving towards presenting the results and discussion together to prevent repetition. That is not a directive, just something to consider. People and journals have different preferences.
Response 12: Thank you for this feedback. We decided to maintain the structure of our paper by keeping the results and discussion sections separate. However, we revised the discussion section with the purpose of reducing redundancies. We deleted many sentences and reduced the paragraph. Red colored sentences are reduced ones.
Comments 13: All things considered, I think this is a very good paper and I commend you on your work.
Response 13: Thank you so much. We appreciate all the feedback you provided to strength our paper.
Reviewer 3 Report
Comments and Suggestions for Authors
1. Abstract
- Keywords: Sort alphabetically.
2. Introduction
- Line 37-38: "There is no national-level research – to our knowledge – conducted on the prevalence and types of IPV among Korean immigrants." Please rearrange this sentence correctly. Is it necessary to use many punctuation marks "–"
- Line 45-47: What exactly does this sentence mean? "The research conducted by Shin [7] found that at least one-third of Korean male respondents reported using physical force against their wives in the previous year". What year was this study conducted? Please clarify the sentence!
- Line 60-64: Is it actually that the sentence refers to references 1 through 4? Kindly check again!
- Why is this study limited to age and gender? Why aren't other factors examined in this study? Kindly explain.
3. Materials and Methods
- Lines 125–125: Table 1. Three categories comprise the Primary language section: Korean, English, and Both. There is 0% of the English category filled. But given that no one uses English, why are seven people in both categories?
4. Discussion
- Lines 632–655: Could you please include citations?
5. Conclusion:
- Please review the conclusion again to ensure it meets the study objectives.
Author Response
Comments 1: Abstract - Keywords: Sort alphabetically.
Response 1: Thank you for pointing out. We sorted them alphabetically. This change can be found at the page 1, line 32-33: “Confucian culture, domestic violence, family violence, focus group, immigrant women, intimate partner violence, Korean American, Korean culture, Korean immigrant”
Comments 2: - Line 37-38: "There is no national-level research – to our knowledge – conducted on the prevalence and types of IPV among Korean immigrants." Please rearrange this sentence correctly. Is it necessary to use many punctuation marks "–"
Response 2: Thank you for your feedback. We agree with you. We made the change, and it can be found at page 1, line 37-39: “To our knowledge, there is no national-level research conducted on the prevalence and types of IPV among Korean immigrants.”
Comments 3: Line 45-47: What exactly does this sentence mean? "The research conducted by Shin [7] found that at least one-third of Korean male respondents reported using physical force against their wives in the previous year". What year was this study conducted? Please clarify the sentence!
Response 3: Thank you for your feedback. We added what year it was. The change can be found at the page 2, line 47: “The research conducted by Shin (Shin, 1995) found that at least one-third of Korean male respondents reported using physical force against their wives in the past 12 months prior to the study conducted in 1995”
Comments 4: Line 60-64: Is it actually that the sentence refers to references 1 through 4? Kindly check again!
Response 4: Thank you for pointing that out. We rearranged the barriers and rewrote the sentence again in a way that is clear, instead of being mixed with other articles. The change can be found at page 2, line 61-65: “Unique barriers for immigrant survivors include immigration status, social isolation and lack of family support, cultural and religious barriers, language and financial barriers, limited awareness of available resources, limited use and perceived inefficacy of services, fear of being deported and separated from their children and psychological barriers such as a feeling of shame and stigma, etc. [1].”
Comments 5: Why is this study limited to age and gender? Why aren't other factors examined in this study? Kindly explain.
Response 5: Thank you for asking a very important question. Previous studies have highlighted that age and gender are significant factors influencing community members' knowledge, attitudes, and behaviors toward IPV (Álvarez, Aranda, & Huerto, 2015; Han, Jeong, & Kim, 2017). Our study focused on these dimensions to identify potential differences among participants that could inform the development of targeted community-wide interventions based on these demographic distinctions.
- Álvarez, C. D., Aranda, B. E., & Huerto, J. A. L. (2015). Gender and cultural effects on perception of psychological violence in the partner. Psicothema, 27(4), 381-387.
- Han, Y. R., Jeong, G. H., & Kim, S. J. (2017). Factors influencing beliefs about intimate partner violence among adults in S outh K orea. Public health nursing, 34(5), 412-421.
Comments 6: Lines 125–125: Table 1. Three categories comprise the Primary language section: Korean, English, and Both. There is 0% of the English category filled. But given that no one uses English, why are seven people in both categories?
Response 6: Thank you for asking. None of the participants used solely English as their primary language. However, seven participants are bilingual and use both English and Korean in their daily lives as primary languages.
Comments 7: Lines 632–655: Could you please include citations?
Response 7: Thank you for your feedback. We added more citations that could strengthen the arguments. Additionally, we rewrote sentences that could have included additional evidence. The change can be found at the page 14, citation number 44,45,29,46, line 652-669.
Comments 8: Please review the conclusion again to ensure it meets the study objectives.
Response 8: Thank you so much for your feedback. We made a significant change in our conclusion. The change can be found at Page 15, line 727-730, line 740-747: “This study sheds light on Korean immigrants’ perceptions of IPV and cultural and practical barriers to seeking help, with an emphasis on age and gender. The findings highlight significant disparities, most notably between older men and other demographic groups. Practical barriers such as … By identifying community perceptions of IPV and barriers to help seeking, the study underscores the need for culturally sensitive and age, gender specific interventions that address the unique needs of immigrant communities. … The study provides a foundation for future research and intervention strategies aimed at changing community perceptions and attitudes toward IPV in a way that improves the help-seeking behaviors and… “